# Precision Medicine Tumor Boards: Clinical Applicability of Personalized Treatment Concepts in Ovarian Cancer

**DOI:** 10.3390/cancers12030548

**Published:** 2020-02-27

**Authors:** Stefanie Aust, Richard Schwameis, Tamara Gagic, Leonhard Müllauer, Eva Langthaler, Gerald Prager, Christina Grech, Alexander Reinthaller, Michael Krainer, Dietmar Pils, Christoph Grimm, Stephan Polterauer

**Affiliations:** 1Department of General Gynecology and Gynecologic Oncology, Gynecologic Cancer Unit, Comprehensive Cancer Center, Medical University of Vienna, 1090 Vienna, Austria; stefanie.aust@meduniwien.ac.at (S.A.); richard.schwameis@meduniwien.ac.at (R.S.); TamaraGagic@msn.com (T.G.); christina.grech@meduniwien.ac.at (C.G.); alexander.reinthaller@meduniwien.ac.at (A.R.); stephan.polterauer@meduniwien.ac.at (S.P.); 2Department of Pathology, Medical University of Vienna, 1090 Vienna, Austria; leonhard.muellauer@meduniwien.ac.at (L.M.); eva.langthaler@meduniwien.ac.at (E.L.); 3Clinical Division of Oncology, Department of Medicine I, Comprehensive Cancer Center, Medical University of Vienna, 1090 Vienna, Austria; gerald.prager@meduniwien.ac.at (G.P.); michael.krainer@meduniwien.ac.at (M.K.); 4Division of General Surgery, Department of Surgery, Comprehensive Cancer Center Vienna (CCC) Vienna, Medical University of Vienna, 1090 Vienna, Austria; dietmar.pils@univie.ac.at

**Keywords:** precision medicine tumor board, ovarian cancer, targeted treatment

## Abstract

Background: Treating cancer according to its molecular alterations (i.e., targeted treatment, TT) is the goal of precision medicine tumor boards (PTBs). Their clinical applicability has been evaluated for ovarian cancer patients in this analysis. Methods: All consecutive ovarian cancer patients discussed in a PTB at the Medical University of Vienna, Austria, from April 2015 to April 2019 were included (*n* = 44). Results: In 38/44 (86%) cases, at least one mutation, deletion or amplification was detected. The most frequently altered genes were p53 (64%), PI3K pathway (18%), KRAS (14%), BRCA1 (11%) and BRCA2 (2%). In 31 patients (70%) a TT was recommended. A total of 12/31 patients (39%) received the recommended therapy. Median time from indication for PTB to TT start was 65 days (15–216). Median time to treatment failure was 2.7 months (0.2–13.2). Clinical benefit rate (CBR) was 42%. Reasons for treatment discontinuation were disease progression (42%), poor performance status (PS > 2; 25%), death (17%) or treatment related side effects (8%). In 61% the TT was not administered—mainly due to PS > 2. Conclusion: Even though a TT recommendation can be derived frequently, clinical applicability remains limited due to poor patients’ general condition after exploitation of standard treatment. However, we observed antitumor activity in a substantial number of heavily pretreated patients.

## 1. Introduction

The management of ovarian cancer has emerged from a “one size fits all” strategy to therapy guidelines according to histological subtypes, differentiating management of epithelial and non-epithelial, as well as common and rare ovarian tumors [1,2,3]. Even though molecular signatures stratifying patients further into biologically different subtypes (differentiated, immuno-reactive, mesenchymal and proliferative) have been described and validated, clinical applicability could not be established so far [4,5,6,7]. The growing knowledge of defects in homologous recombination repair in ovarian cancer has finally led to the successful implementation of PARP inhibitors in standard treatment [8,9]. Besides bevacizumab and PARP inhibitors, the development of targeted therapies based on molecular markers faces several challenges in ovarian cancer.

The success of combining molecular analyses and targeted treatments has been proven in several cancer entities. The HER2 antibody trastuzumab [10] or lapatinib, a dual EGFR-erbB2 TKI, as well as other tyrosine kinase inhibitors [11] in breast cancer, the epidermal growth factor inhibitors gefitinib and erlotinib in non-small cell lung cancer [12,13] or the BRAF inhibitor vemurafenib in melanoma [14] exemplify the efficacy of targeted agents.

Precision medicine is an approach to identify targetable molecular characteristics of a tumor beyond the borders of histological tumor origin, followed by a respective targeted treatment to control that tumor and thereby prolong survival. Molecular aberrations involved in cancer development, progression, recurrence and resistance are the fundament of those targeted therapies.

As outcome analyses regarding “molecular” or “precision medicine” tumor boards (PTBs) are evolving [15,16], there are several challenges related to an integration of precision medicine into clinical routine [17,18,19]. These comprise biomarker definition, the cost of molecular analyses, fundamental molecular biological knowhow, implementation of expert guided molecular tumor boards and optimization of clinical applicability of potential targeted therapies (TTs), followed by outcome and adverse event reporting as well as their validation [20].

The success of PARP inhibitors shows that targeted treatment is possible in ovarian cancer. Still, several questions need to be discussed critically [21]. Targeted therapies are usually studied in metastatic disease after failure of standard treatment. But metastatic and therapy refractory solid cancers are heterogeneous, present with accumulated genetic derangements [22] and are challenging to treat effectively.

The purpose of this study was thus to evaluate the clinical applicability and utility as well as the current limits of PTBs for ovarian cancer patients in a university hospital setting. To achieve this purpose, we analyzed the clinical data (tumor and patient characteristics at diagnosis and time of indication of TTs, time to treatment failure of TTs and outcome parameters) of all ovarian cancer patients that underwent genomic profiling and respective integration in a multidisciplinary PTB platform [23] in the Medical University of Vienna, Austria, over a four-year period.

We also aimed at addressing practical and logistical issues, such as time from indication of PTB to actual TT start and the challenges associated with implementing TTs for patients with advanced ovarian cancer.

## 2. Results

Between April 2015 and April 2019, 44 ovarian cancer patients were discussed in an interdisciplinary PTB at the Medical University of Vienna, Austria. The patient group included different histological subtypes of ovarian cancer (Table 1). The molecular profile was determined after up to six lines (range one to six lines) of previous systemic therapy (median three previous therapies per patient). Five patients had already received a PARP inhibitor treatment previous to the PTB. At the time of data evaluation, 29 (66%) patients had died, 14 (32%) were still alive and one patient was lost to follow up.

The respective tumor tissue was obtained via interventional radiological techniques or by surgery (i.e., lymph node excision, surgery in the course of mechanical complications such as occlusion and intraabdominal pain, explorative surgery and tumor excision). In total, 18 samples were from metastatic and 25 from local/intraperitoneal lesions. In one patient, description of sample origin was not documented. Origin of tumor tissue obtained by real-time biopsies was: Lymph nodes (*n* = 7), liver (*n* = 6), vagina (*n* = 2), breast (*n* = 2) and lung (*n* = 1).

### 2.1. Genomic Findings

In 86% of patients (*n* = 38), at least one mutation could be identified. Thereof, the mean number of mutations per patient was 1.74. A maximum of six mutations was present in one patient with endometrioid ovarian carcinoma (MSH6, PIK3CA, FBXW7, PIK3R1, PITCH1, ERBB3 and PPP2R1A). No mutations were found in three HGSC patients, one mucinous adenocarcinoma and two malignant sex cord-stromal cell tumors.

Mutations in 22 different genes were identified (whole gene-panel depicted in methods). TP53 was mutated in 64% (*n* = 28). Headed by TP53, a total of five genes were altered repeatedly (Figure 1).

In the sub-group of HGSC (*n* = 31), mutations were detected repeatedly in TP53 (81%; *n* = 25), BRCA 1 (16%; *n* = 5), PIK3CA (10%; *n* = 3) and PIK3R1 (6%; *n* = 2), followed by mutations in KRAS, MET, RET, KIT, CDH1, FBXW7, ATM, NF1 and TERT, detected each in one patient only.

Comparing the results obtained within different tumor tissues shows that at least one mutation was detected in all 18 metastatic samples and in 77% of local/intraperitoneal samples (in six out of the 26 intraperitoneal samples no mutation was found; *p* = 0.067, Fisher’s exact test).

#### Simultaneous Mutations in Targetable Genes

Within the five patients harboring a PIK3CA mutation, a simultaneous KRAS mutation was present in one patient, in two patients a simultaneous BRCA mutation was detected (BRCA 1, *n* = 1 and BRCA2, *n* = 1) and in one patient a simultaneous MSH6 mutation.

### 2.2. Immunohistochemical Findings

IHC could not be analyzed in two patients due to insufficient tumor tissue. Of the remaining 42 samples, EGFR expression was positive in 34 (81%) with a median score of 150 (range: 15–280) and strong in 10 (24%) samples. P-mTOR was expressed in 38 samples (90%) with a median score of 150 (range: 30–300) and high expression in 11 (26%) samples. PTEN expression was negative in four (9%) samples. A total of 31 samples were hormone receptor positive—29 (69%) ER, 20 (48%) PR and 18 (43%) both ER and PR positive. PD-L1 positive tumor cells were present in eight (19%) and a combined positive score of ≥5 in 12 (29%) samples.

### 2.3. Treatment Allocation

In 31 patients (70% of the total patient group) a TT was proposed (Table 2) based on the tumor’s molecular profile. In three/31 patients (10%) a TT was recommended against potentially actionable mutations and in 28/31 patients (90%) according to immunohistochemically determined expression patterns and patient characteristics. Three patients harboring a BRCA mutation had already received a TT (PARP inhibitor) previously within clinical routine.

### 2.4. Clinical Endpoints

Out of the 31 patients with a TT recommendation, 39% (*n* = 12) received the respective therapy: PI3K-AKT/mTOR inhibitor combined with antiestrogen therapy (*n* = 5); immune checkpoint inhibitor (ICI; *n* = 4); PARP-inhibitor (*n* = 2); sunitinib together with an aromatase inhibitor (*n* = 1).

Median time to treatment failure (TTF; Figure 2) was 2.7 months (range 0.2 to 13.2 months; in one patient treatment was still ongoing at time of data evaluation). Clinical benefit rate (CBR) at three months was 42% (complete response, *n* = 0; partial response, *n* = 3; stable disease, *n* = 2). One year overall survival (OS)-rate after PTB meeting was 39%. Characteristics of the 12 patients and reasons for treatment discontinuation are shown in Table 3.

In the subgroup of patients receiving a PI3K-AKT/mTOR inhibitor together with an antiestrogen therapy, the median TTF was 1.1 months (range 0.2 to 10.3 months). In one patient (HGSC, pre-treated with four different chemotherapy regimens) this targeted therapy was administered (together with bevacizumab) for 10.3 months until disease progression. Subsequently, another treatment with carboplatin as monotherapy was reintroduced. At the time of analysis (19 months after the respective PTB) the patient was still alive and responding to chemotherapy.

In the majority of patients (19 out of 31; 61%;) the corresponding TT was not administered. Two patients (6%) had no progressive disease at the time of analysis. One patient (3%) was lost to follow up. In 16 patients (52%) the TT was not given due to an impaired general condition. In detail, 10 patients (29%) experienced a rapid decline in performance status (PS > 2) and were referred to palliative/hospice care or died shortly after the PTB meeting and six patients (19%) continued their still ongoing chemotherapy until disease progression, followed by a decline in performance status that made further implementation of a TT impossible. One year OS-rate after PTB meeting was 24% in these patients (*n* = 19). Within this context, the time from diagnosis of recurrence to start of TT is of particular interest in this cohort of morbid patients in their end stage of disease. The median time between clinical presentation with recurrence (need for alternative treatment) and TT start was 65 days (range 15 to 216).

## 3. Discussion

In this study, implementation of precision medicine tumor board recommendations for ovarian cancer patients was evaluated, focusing on the abundance of targetable aberrations, targeted therapy recommendations and their subsequent clinical applicability and limitations.

Comprehensive genomic profiling to assign TTs has been proven feasible in a subset of heavily pretreated patients with metastatic cancer, including ovarian cancer [24]. Still, already back in 2011, the Cancer Genome Atlas project revealed that the number of detectable driver mutations remains limited in ovarian cancer compared to other cancer entities [25]. Subsequently, the evaluation of targeted therapies related to infrequent molecular markers remains challenging. Taking into account the low incidence of ovarian cancer and the particularly small molecular subsets characterized by mutations in specific genes or expression profiles, and considering that tumors with an annual incidence rate of ≤6 per 100,000 are classified as “rare” [26], molecular subsets with specific rare mutations should be classified accordingly. Reaching the required patient numbers to perform conventional randomized controlled clinical trials—even on an international level—remains difficult [27]. Information obtained in molecular or precision medicine tumor boards is; thus, gaining relevance to stratify patients into genotype-matched clinical trials [28], and subsequently to determine treatment effectivity of targeted therapies for low frequent mutations [27,29].

Continuous development and cost reduction of sequencing techniques enhance the possibility of characterizing a tumor at any given time point [30]. A multi-disciplinary PTB is therefore the optimal platform to understand biological implications of detected mutations and to derive therapeutic consequences if applicable.

Consistent with pre-existing data [31,32], aberrations of the PI3K pathway were the most frequent targetable genetic alterations. PIK3CA mutation was detected in 11%, followed by alterations of the core member PIK3R1 present in 7% of patients and discussed as biomarker of responsiveness if targeting the PI3K pathway [33]. KRAS was mutated in 14%, and associated with lower grade and mucinous subtype, as already previously published [31,34]. Beside TP53 mutations (in total 64%; 81% in HGSC) and BRCA mutations (BRCA 1 = 11%; BRCA 2 = 2%), the majority (66.6%) of detected genetic alterations (CDH1, FOXL2, MET, RET, KIT, MSH6, FBXW7, PPP2R1A, PITCH1, ERBB3, APC, ATM, TERT, NF1, MYCN and ABL1) were mutations of low frequency (<10%) and mutually exclusive, contributing to our understanding of the heterogeneity of ovarian cancer and its rather low mutational burden [35,36].

Given the intra-patient genomic variability, acquired resistance and tumor evolution, the question remains if the origin of analyzed tissue has an impact [37,38]. As we only analyzed one tissue sample per patient, intra-patient variability could not be determined. Comparing the results according to tissue origin, at least one mutation was detected in 100% of metastatic samples obtained by biopsies, compared to 77% of the 26 local/intraperitoneal samples obtained surgically. Interpretation of these results remains limited but suggests that biopsies from metastatic lesions are an equivalent source for molecular analyses in ovarian cancer.

In our PTBs, a TT was recommended in 70% (*n* = 31) of ovarian cancer patients. This rate is slightly higher compared to other reports in gynecologic oncology [35,39], whereby selected data on PTB outcomes in ovarian cancer patients are limited. This can be explained by treatment propositions not only given according to a screening of actionable genomic alterations [40] but also considering the immunohistochemical tumor profile and patient’s history.

Of note, only 27% of ovarian cancer patients (*n* = 12) entering the PTB program received a respective TT. This is in line with previously published data by Spreafico et al. in ovarian cancer [41]. Median TTF was 2.7 months. A continuous therapy efficacy of 10 and 13 months until TTF could be archived in two patients. CBR at three months was 42% (CR *n* = 0, PR *n* = 3, SD = 2).

The question if combining individualized treatment approaches might increase efficacy of targeted therapies in advanced recurrent ovarian cancer remains difficult to determine, as data on applying multitargeted therapeutic approaches is limited. Especially the combination with PI3K-Akt-mTOR inhibitors often requires compromised doses due to dose-limiting toxicities [42]. Combinations are; thus, limited by availability, toxicities [43] and the related question of dosing adaptation.

Can a TT be effective in heavily pre-treated advanced stage ovarian cancer patients [44]? Especially in this type of solid tumors, where an extensive tumor-burden is usually present when an alternative to standard treatment is sought, it has to be admitted that it remains difficult for a molecularly-targeted agent to be sufficiently effective. The authors of the SHIVA trial have likewise questioned the possibility of TTs to be adequately effective in heavily-pretreated patients with solid tumors [45]. Additionally, as stated by Tannock et al. the Darwinian evolution and consequent intratumor heterogeneity might limit the effectiveness of TTs [46].

We are still facing several limitations implementing PTB recommendations in clinical routine. A key limitation is the costs of the underlying assays (sequencing, antibody development) that are required for comprehensive PTBs. Fortunately, costs are constantly decreasing although the panels are covering broader aspects of the tumor biology. This is reflected in our study, as, mainly caused by financial reasons, initially only a 50-gene panel could be analyzed, followed by a 160-gene panel that will soon be replaced by a 500-gene panel.

Additionally, the evaluated study group consisted of heavily-pretreated ovarian cancer patients with up to six lines of previous systemic therapy. In 61% of patients with a TT recommendation, the respective therapy could not be administered, mainly due to rapid disease progression and degradation of patients’ general condition (PS >2; 52%), leaving a TT approach out of consideration. Impaired general status was likewise a limit of access to TTs in large multicenter trials [39]. Thus, it has to be further discussed at what timepoint we should optimally screen for—and propose treatment with—molecularly-targeted agents. The main reasons for entering the PTB program, but not receiving a TT (*n* = 32/44, 73%), were lack of an actionable target (*n* = 13/32, 41%) and impaired performance status at the time of TT start (*n* = 16/32, 50%). A particular risk factor could be the time between diagnosing recurrence and TT start. In our study, median time between indication for PTB and TT was 65 days. Time from recurrence to TT start seems to be particularly important in these patients, as they typically present with impaired performance status, malnutrition, impairing symptoms (ascites, pleural effusion, and ileus) and/or progressive disease. Thus, a concise algorithm for the PTB to shorten this time as much as possible seems to be of utmost importance.

## 4. Material and Methods

### 4.1. Patient Selection

All consecutive patients with recurrent, (standard) therapy refractory ovarian cancer treated at the Department of Gynecology and Gynecological Oncology at the Medical University of Vienna, Austria, who were discussed in our PTB [23], were included in the study. Inclusion criteria comprised informed consent, real-time tumor biopsy or tumor tissue samples derived from surgery, Eastern Cooperative Oncology Group performance status (PS) ≤ 1, life expectancy of more than 3 months and patient ambition of treatment continuation. Patient flow-chart is given in Figure 3.

### 4.2. Tumor Tissue Analyses

Tumor DNA was extracted from formalin-fixed, paraffin-embedded tissue from tissue samples obtained before the PTB. The most recent material available was used. Two molecular profiling assays were used over the study period. Until June 2018 The DNA library was generated by multiplex polymerase chain reaction with the Ion AmpliSeq Cancer Hotspot Panel v2TM (Life Technologies, Carlsbad, CA, USA), enabling detection of mutation hotspots of the following 50 genes: ABL, AKT, ALK, APC, ATM, BRAF, CDH, CDKN2A, CSF1R, CTNNB1, EGFR, ERBB2, ERBB4, EZH2, FBXW7, FGFR1, FGFR2, FGFR3, FLT3, GNA11, GNAS, GNAQ, HNF1A, HRAS, IDH1, JAK2, JAK3, IDH2, KDR, KIT, KRAS, MET, MLH1, MPL, NOTCH1, NPM1, NRAS, PDGFRA, PIK3CA, PTEN, PTPN11, RB1, RET, SMAD4, SMARCB1, SMO, SRC, STK11, TP53, VHL, BRCA 1 and BRCA 2. Molecular profiling was additionally performed for each patient using the Ion AmpliSeq BRCA1 and BRCA2 Panel (Thermo Fisher, Waltham, MA, USA). In June 2018 we changed to the Oncomine Comprehensive Assay v3, allowing a detection of variants across 161 genes (AKT1, ALK, AR, ARAF, BRAF, BTK, CBL, CDK4, CHEK2, CSF1R, CTNNB1, DDR2, EGFR, ERBB2, ERB83, ERBB4, ESR1, EZH2, FGFR1, FGFR2, FGFR3, FLT3, FOXL2, GATA2, GNA11, GNAQ, GNAS, HNF1A, HRAS, IDH1, IDH2, JAK1, JAK2, JAK3, KDR, KIT, KNSTRN, KRAS, MAGOH, MAP2K1, MAP2K2, MAPK1, MAX, MED12, MET, MTOR, MYD88, NFE2L2, NRAS, PDGFRA, PIK3CA, PPP2R1A, PTPN11, RAC1, RAF1, RET, RHEB, RHOA, SF3B1, SMO, SPOP, SRC, STAT3,U2AF1, XPO1, AKT2, AKT3, AXL, CCND1, CDK6, ERCC2, FGFR4, H3F3A, HIST1H3B, MAP2K4, MDM4, MYC, MYCN, NTRK1, NTRK2, PDGFRB, PIK3CB, ROS1, SMAD4, TERT, TOP1, ATM, BAP1, BRCA1, BRCA2, CDKN2A, FBXW7, MSH2, NF1, NF2, NOTCH1, PIK3R1, PTCH1, PTEN, RB1, SMARCB1, STK11, TP53, TSC1, TSC2, ARID1A, ATR, ATRX, CDK12, CDKN1B, CDKN2B, CHEK1, CREBBP, FANCA, FANCD2, FANCI, MLH1, MRE11A, MSH6, NBN, NOTCH2, NOTCH3, PALB2, PMS2, POLE, RAD50, RAD51, RAD51B, RAD51C, RAD51DRNF43, SETD2, SLX4 and SMARCA4). Until June 2018, 27 patients had been tested with the Ion AmpliSeq Cancer Hotspot Panel v2TM (+BRCA1 and BRCA2 Panel), followed by 17 patients tested with the Oncomine Comprehensive Assay v3. Sequences were mapped to the human hg19 reference genome. Sequence variants were annotated by comparison with gene sequences deposited in the databases 1000 genomes, 5000 exomes (Exac, UCSC common SNPs and dbSNP). Only sequence variants that were present in these databases at a minor allele frequency below 1%, or not recorded at all, were further annotated by comparison with sequence variants deposited and classified in ClinVar, COSMIC and BRCA Exchange databases. The sequence variants were then classified as “pathogenic/likely pathogenic” or “of uncertain significance”. Benign/likely benign mutations and variants/polymorphisms were not included in the final molecular pathology report.

Immunohistochemistry (IHC) was performed at the Department of Pathology at the Medical University of Vienna using a Ventana Benchmark Ultra Stainer (Ventana, Tucson, AZ, USA). Applying institutionally-approved, standardized procedures, the following antibodies were integrated in this analysis: EGFR (clone 3C6; Ventana), MET (clone SP44; Ventana), estrogen-receptor (ER) (clone SP1; Ventana), progesterone-receptor (PR) (clone 1E2; Ventana), HER2 (clone 4B5; Ventana), phospho-mTOR (clone 49F9; Cell Signaling, Danvers, MS), PD-L1 (clone E1L3N; Cell Signaling; Clone BSR90; Nordic Biosite, Täby, Sweden), and PTEN (clone Y184; Abcam). Staining results were categorized by IHC scoring (multiplying the percentage of positive cells by their respective staining intensity, maximum score 300; for EGFR, phosphor-mTOR/mTOR and PTEN), by percentage of positive tumor cells (PD-L1, ER, PR), combined positive score (CPS; PD-L1), or trichotomized according to established scoring systems (MET).

### 4.3. Outcome Evaluation

TTF was defined as the time from treatment start to discontinuation for any reason, including reduced PS, disease progression, treatment toxicity or death. Even though TTF is an endpoint influenced by factors unrelated to efficacy, it reflects well the clinical applicability of targeted therapies in heavily-pretreated patients [24]. CBR was defined as percentage of patients with complete response, partial response, or stable disease at first evaluation three months after TT initiation. Evaluation was performed by RECIST 1.1 criteria.

## 5. Conclusions

It remains important to discuss the current limitations of clinical applicability of PTBs to further develop strategies to increase their effectiveness and significance. This single center experience highlights the need for a quick implementation of targeted therapies in ovarian cancer patients. We could show that, due to the poor general condition of ovarian cancer patients after exploitation of standard treatment and a prolonged time between need for alternative treatment and TT initiation, patient eligibility decreases substantially. Looking up to the model of breast cancer, where the therapeutic relevance of molecularly classifying each individual cancer has led to various therapy adaptations and, consequently, significant outcome improvements, we believe that the knowledge generated in PTBs will continuously contribute to progress in drug development based on molecular profiles also in ovarian cancer.

## Figures and Tables

**Figure 1 cancers-12-00548-f001:**
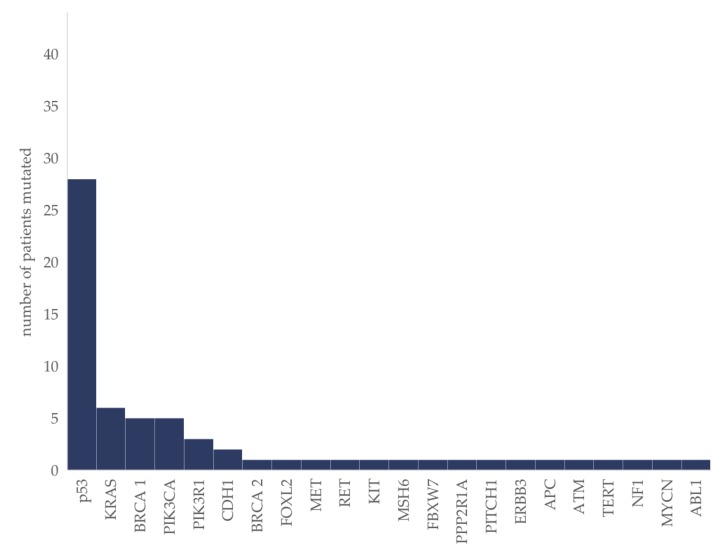
Molecular aberrations in ovarian cancer patients (*n* = 44). The absolute number of patients in which the respective mutation was identified is given.

**Figure 2 cancers-12-00548-f002:**
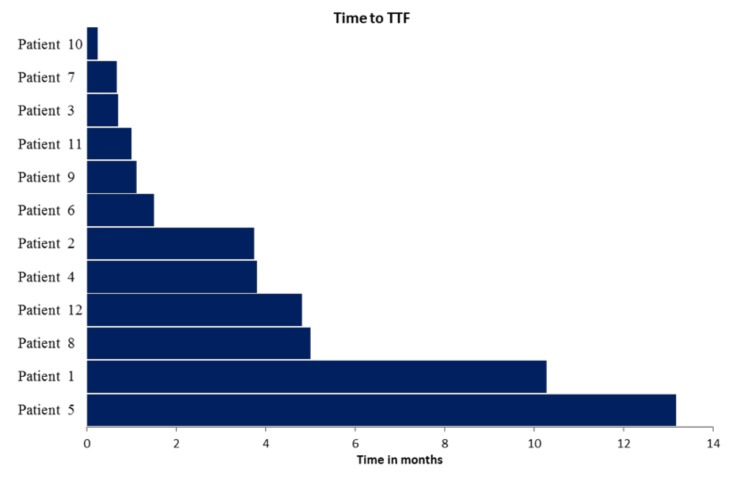
Time to treatment failure (TTF) in the 12 patients who received PTB recommended treatment. Respective patient characteristics and targeted treatment are depicted in Table 2.

**Figure 3 cancers-12-00548-f003:**
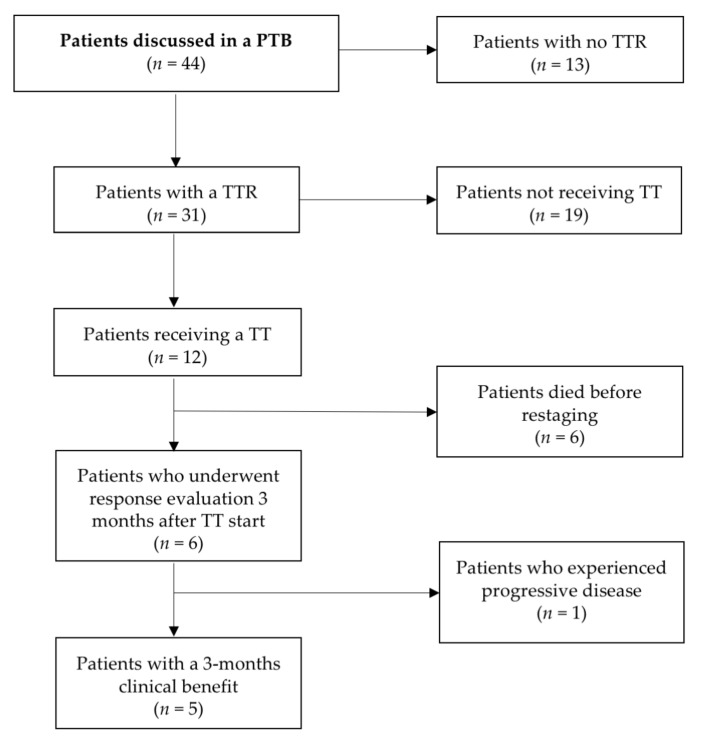
Patient flow-chart.

**Table 1 cancers-12-00548-t001:** Patient characteristics.

Histological Subtype	*n* (%)	Age at Diagnosis Median (Range)	Previous Therapies	Age at PTB Median (Range)
Serous adenocarcinoma	32 (73%)	52 (28–75)	1–6	57 (31–78)
LGSC (*n* = 1) *				
HGSC (*n* = 31)				
FIGO I + II: *n* = 2				
FIGO III + IV: *n* = 30				
Mucinous adenocarcinoma	5 (11%)	51 (40-59)	1–3	53 (43–62)
Endometrioid adenocarcinoma	2 (5%)	61 (49–72)	1	64 (60–77)
Yolc sac tumor	1 (2%)	49	4	50
SCST ^1^	3 (7%)	36 (32–40)	3	48 (34–70)
Carcinosarcoma	1 (2%)	66	4	68

^1^ SCST = Malignant sex cord-stromal cell tumors; * serous borderline tumor that had evolved into a low-grade serous cancer (LGSC); high-grade serous cancer (HGSC); Fédération Internationale de Gynécologie et d’Obstétrique (FIGO), precision medicine tumor board (PTB).

**Table 2 cancers-12-00548-t002:** Recommended targeted therapies.

Targeted Treatment Recommendation	*n*/44 (%)
Everolimus + antiestrogen therapy	14 (32%)
Immune-checkpoint inhibitor	5 (11%)
Everolimus	3 (7%)
Aromatase inhibitor	5 (11%)
PARP inhibitor	2 (5%)
Cetuximab	1 (2%)
Sunitinib + aromatase inhibitor	1 (2%)

**Table 3 cancers-12-00548-t003:** Characteristics of the 12 patients receiving a targeted treatment based on the precision medicine tumor board (PTB) recommendation.

Nr	Histology	Mutation	Deletion/Amplification	VUS	IHC	Prev. Th (*n*)	Targeted Therapy	Age at PTB	TTF (m)	Response Eval.°	End of Th. (cause)	Status	Survival After PTB (m)
1	HGSC	CDH1, TP53	Deletion: PIK3R1, APC, SMAD4		EGFR score = 80, ER 90% (Allred = 8), PR 2% (Allred = 3), PTEN score = 140, mTOR score = 125, PDL1 = neg	4	Everolimus + Letrozol + Bevacicumab	53.7	10.3	SD	PD	alive	18.8
2	HGSC	TP53, BRCA1	Amplification: PPARG, PIK3CA, MET, BRAF	RAD50	ER 100% (Allred = 8), PDL1 = pos (TPS = 1; CPS = 5), PTEN score = 140, mTOR score = 110, MET score = 30	6	Nivolumab	49.0	3.7	PD	PD	alive	10.8
3	HGSC	TP53			EGFR score = 260, PR pos, ER = neg, PTEN score = 110, p-mTOR score = 30, PDL1 pos (TPS = 50, CPS = 60)	2	Pembrolizumab + Bevacizumab	54.4	0.7	n.a.	death	death	0.8
4	HGSC	KRAS		FGFR1	EGFR score = 200, mTOR = score 90, PTEN score = 100, PDL1 = pos (TPS = 0, CPS = 5)	5	Rucaparib	60.7	3.8 *	PR	n.a.*	alive	3.8
5	HGSC	BRCA1, TP53, PIK3CA	Amplification: FGFR1, NTRK3, IGF1R	POLE (Eon32), POLE (Eon6), NOTCH3	EGFR score = 180, MET low, ER 80% (Allred:8), PR 5% (Allred = 5), PTEN = pos, p-mTOR score = 140, PDL1 = neg	2	Olaparib	58.8	13.2	PR	PD	death	32.3
6	HGSC	TP53			EGFR neg, MET low, ER = 90% (Allred = 7), PTEN = n.d., PD1-Ligand = faible pos, PDGFRalpha = moderately pos	6	Anastrozole + Sunitinib	55.2	1.7	n.a.	PS > 2, toxic liver failure	death	2.2
7	HGSC	ATM, PIK3R1, TP53			EGFR = high (Score:210), ER = 100% (Allred = 8), PR = > 90% (Allred = 8), PDL1 = TPS = 1, CPS = 15; p-mTOR score = 270, PTEN score = 70	5	Everolimus + Exemestan	56.3	0.7	n.a.	death	death	1.1
8	HGSC	TP53			EGFR = neg, MET = low, ER = 80% (Allred = 8), PR = 95% (Allred = 8), PTEN = neg, p-mTOR score = 70, PDL1 = neg	1	Everolimus + Exemestan	63.4	5.0	PR	PD	death	17.4
9	HGSC	TP53			EGFR score = 20s, ER = 80% (Allred = 8), PR = 15%(Allred = 6), p-mTOR score = 300, PDL1 = neg, MET = neg, PTEN = neg	1	Everolimus + Exemestan	64.7	1.1	n.a.	PS > 2	death	1.4
10	HGSC	PIK3R1, TP53, NF1			EGFR = neg, ER = 80% (Allred = 7), PTEN score = 150, mTOR score = 180, PDL1 = neg	4	Everolimus + Exemestan	55.1	0.2	n.a.	PS > 2	death	1.3
11	HGSC	PIK3CA			EGFR score = 210, ER = 20 (Allred = 6), PTEN score = 90, p-mTOR score = 45, PDL1 = pos (TPS 50, CPS 50)	2	Avelumab	39.8	1.0	n.a.	treatment related side effects	death	6.0
12	LGSC	KRAS, ABL1			EGFR score =70, MET = neg, ER = 90% (Allred = 8), PR = 40% (Allred = 5), PTEN score = 200, p-m-TOR score = 290, PDL1 pos (TPS = 2, CPS = 50)	6	Pembrolizumab	48.6	4.8	SD	PD	death	5.0

* Therapy still ongoing; ° response at three months; n.a., not applicable; n.d., not done; VUS, variant of unclear significance; IHC, immunohistochemistry; Prev.Th, number of previous therapies; TTF, time to treatment failure; m, months; PS, performance status; PD, progressive disease; PTB, precision medicine tumor board; HGSC, high-grade serous cancer; LGSC, low-grade serous cancer; KRAS, kirsten rat sarcoma viral oncogene; BRCA, breast cancer gene; PIK3CA, Phosphatidylinositol-4,5-Bisphosphate 3-Kinase Catalytic Subunit Alpha; PIK3R1, Phosphoinositide-3-Kinase Regulatory Subunit; CDH1, Cadherin-1; MET, proto-oncogene MET; APC, adenomatous-polyposis-coli gene; NF1, neurofibromin; ABL1, abelson murine leukemia viral oncogene homolog 1; FGFR1, fibroblast growth factor receptor 1, NTRK3, neurotrophic tyrosine receptor kinase, IGF1R, insulin-like growth factor 1 receptor, SMAD4, SMAD family member 4; BRAF, v-Raf murine sarcoma viral oncogene homolog B1; PPARG, peroxisome proliferator-activated receptor gamma, RAD50, DNA repair protein RAD50; POLE, DNA polymerase epsilon, catalytic subunit; NOTCH, notch receptor; EGFR, epidermal growth factor receptor; ER, estrogen receptor; PR, progesterone receptor; mTOR, mechanistic target of rapamycin; PTEN, phosphatase and tensin homolog; PDL1, programmed death-ligand 1; CPS, combined prognostic score; PDGFR, platelet-derived growth factor receptor.

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
