# Peer review of "Precision Medicine Tumor Boards: Clinical Applicability of Personalized Treatment Concepts in Ovarian Cancer"

_cancers, 2020, doi:10.3390/cancers12030548_

Round 1

Reviewer 1 Report

The authors replied very well to all comments and questions, improving the paper. They reviewed most cases updating table 1. I would recommend to accept this paper.

Only one very minor remark: in the abstract, the abbrevation CBR is used without explanation (clinical benefit rate).

Author Response

Dear Reviewer, 

thank you very much for your final comments. 

Your minor remark was addressed and the abbreviation CBR was written-out in the abstract. 

Reviewer 2 Report

The revised manuscript is now clearly written. However, according to my practice of describing patients in a study, one sentence needs improvement:

Material and methods, 6.1. Patient selection, lines: 257-261. This sentence is a little bit too long and challenging to read, it should be improoved.

Author Response

We thank the reviewer for his final comments. 

The minor remark was addressed and the sentence in the material and methods section rephrased.

Reviewer 3 Report

Overall, the manuscript is well written. The findings are significant and will be valuable to the concerned scientific community. The authors have revised the manuscript accordingly the reviewers reccomendations. The manuscript is suitable for publication.

Author Response

We thank the reviewer for this positive remark.

Reviewer 4 Report

The manuscript deals with a highly important topic of systemic cancer therapy directed by gene sequence/expression data. Selection of ovarian cancer, and specifically high-grade serous ovarian carcinoma (HGSOC), is highly relevant due to frequent disease recurrence and development of therapeutic resistance to the conventional combination chemotherapy. On the other hand, selection of this disease makes the project more challenging due to the fact that, besides TP53 and BRAC1, there are no other cancer relevant genes that would display recurrent point mutations among HGSOC cases. Instead, the disease had been shown to display the high prevalence of copy number alterations in cancer genome. In addition, some gene alterations in ovarian cancers have not provided comparable therapeutic benefit as in other cancers (e.g. very modest response to trastuzumab or pertuzumab monotherapy in HER2-positive ovarian cancers). 

Due to difficulties limiting inclusion of a more substantial number of patients, this study reports results produced from a limited sample size. The study has not employed any comparison group (parallel or historical), which would have allowed more robust estimate of the significance of reported therapeutic response. Another approach that could have been  considered without control group is time series with statistical analysis of a a quantitative measure of tumor burden, such as CA-125 at the beginning of treatment and another fixed time point. This reviewer would recommend (i) not calculating % from small numbers but rather showing numerators and denominators, (ii) more illustrative presentation of selection of patients, (iii) information on which types of mutations were considered as relevant (any sequence variants? pathogenic variants? probably pathogenic variants? by which model or consensus?), and (iv) more detailed information on the response criteria implemented in this study (complete, partial, stable disease etc) with reference to RECIST or other defined criteria. Other suggestions are included as comments directly in the text of the manuscript.

Author Response

Response to Reviewer IV

The manuscript deals with a highly important topic of systemic cancer therapy directed by gene sequence/expression data. Selection of ovarian cancer, and specifically high-grade serous ovarian carcinoma (HGSOC), is highly relevant due to frequent disease recurrence and development of therapeutic resistance to the conventional combination chemotherapy. On the other hand, selection of this disease makes the project more challenging due to the fact that, besides TP53 and BRAC1, there are no other cancer relevant genes that would display recurrent point mutations among HGSOC cases. Instead, the disease had been shown to display the high prevalence of copy number alterations in cancer genome. In addition, some gene alterations in ovarian cancers have not provided comparable therapeutic benefit as in other cancers (e.g. very modest response to trastuzumab or pertuzumab monotherapy in HER2-positive ovarian cancers). 

Due to difficulties limiting inclusion of a more substantial number of patients, this study reports results produced from a limited sample size. The study has not employed any comparison group (parallel or historical), which would have allowed more robust estimate of the significance of reported therapeutic response. Another approach that could have been  considered without control group is time series with statistical analysis of a a quantitative measure of tumor burden, such as CA-125 at the beginning of treatment and another fixed time point.

This reviewer would recommend (i) not calculating % from small numbers but rather showing numerators and denominators, (ii) more illustrative presentation of selection of patients, (iii) information on which types of mutations were considered as relevant (any sequence variants? pathogenic variants? probably pathogenic variants? by which model or consensus?), and (iv) more detailed information on the response criteria implemented in this study (complete, partial, stable disease etc) with reference to RECIST or other defined criteria. Other suggestions are included as comments directly in the text of the manuscript.

We thank the reviewer for his review and the comments in the manuscript.

We addressed the remarks added by the reviewer and want to highlight the following changes:

(i) Figure 1 was adapted and now the patient numbers not percentages are given;

(ii) more illustrative patient presentation was given in the newly added figure 3 (patient flow-chart)

(iv) response criteria were added in the material and methods section

(v) OS rate at 12 months was added for the group of patients receiving TT and the group not receiving TT. Survival data were not compared directly as differences in survival could have been affected particularly by PS and other baseline characteristics.

(iii) We would like to provide the following answer to the reviewer (please also see reference 23 for more details):

What was considered as a mutation?

Sequences were mapped to the human hg19 reference genome. Sequence variants were annotated by comparison with gene sequences deposited in the databases 1000 genomes, 5000 exomes, Exac, UCSC common SNPs and dbSNP. Only sequence variants that were present in these databases at a minor allele frequency (MAF) below 1% or not recorded at all were further annotated by comparison with sequence variants deposited and classified in ClinVar, COSMIC and BRCA Exchange databases. The sequence variants were then classified as pathogenic/likely pathogenic or "of uncertain significance". Benign/likely benign mutations and variants/polymorphisms were not included in the final molecular pathology report. (this paragraph was added to the methods section)  

Was it any sequence variant in the sequenced genes?

No, only sequence variants that were annotated pathogenic/likely pathogenic or "of uncertain significance" were included in the pathology reports (targeted treatment decisions were only based on pathogenic/likely pathogenic mutations/sequence variants?).

Or any deleterious sequence variant? If deleterious - how was the deleterious status evaluated? By classification of ACMG/AMP?

Nonsense and frameshift mutations and mutations in highly conserved splice-site positions were were regarded as deleterious/likely deleterious (pathogenic/likely pathogenic). Missense mutations were considered as deleterious/likely deleterious only in cases with sufficient support for a functional impairement of the encoded protein by sequence entries in COSMIC, ClinVar or BRCA Exchange and/or supporting reports in the scientific literature.

Reviewer 5 Report

The authors have presented an important study describing the feasibility of Precision Medicine in Ovarian cancer patients. In spite of the limitations which is unavoidable when a study involves cancer patients who are at various stages of disease and have undergone different lines of treatment, the study indicates that there is a sign of hope for ovarian cancer patients.

This is a revised version and I see that majority of queries have been addressed adequately, I have some minor comments:

Please include  abbreviation for CBR in the abstract, like you have done for other terminologies.  Line 42, und ?  Line 64-65 : 64  "Still,several questions, such as the question of timing need to be discussed critically" Please re-phrase this sentence. Line 224-225 : Please re-phrase the paragraph. It sounds vague and redundant.

Author Response

We thank the reviewer for his comments. 

The minor remarks were addressed and the abbreviation CBR was written-out in the abstract. 

Round 2

Reviewer 4 Report

The authors have implemented the recommendations and the manuscript has substantially improved. 

This manuscript is a resubmission of an earlier submission. The following is a list of the peer review reports and author responses from that submission.

Round 1

Reviewer 1 Report

Targeted treatment and personalised medicine are hot topics in current medical practice. Like in all hospitals, molecular profiling of a tumor in order to find a targeted genomic variant is now easily to perform at a relative low cost. However, this molecular testing does not always leads to a change in therapy. In their article, the authors tried to give an overview on the targeted approach of ovarian cancer treatment in their institution. The intention for targeted treatment might be present, but before it leads to  personalised medicine other characteristics such as individual patient performance status and condition need to be taken in account. The paper is clinically valuable as many medical centers struggle with similar problems regarding patient accrual and determining treatment efficacy.

I have several important remarks about the presentation of the results (including the abstract).

Comprehensive genomic profiling was performed – and according to M&M two assays were used: how many patients were tested by the 50 genes Hotspot Panel and how many by the 161 genes Panel? If the majority of patients has only been tested by the 50 genes Panel, it is not really “comprehensive profiling”. The molecular profile was determined after up to 6 lines of previous systemic therapy: was this the same for all patients? How many patients did already receive PARP-inhibitors which is actually some sort of TT? Table 1. Since the WHO classification of 2014 (which will be updated beginning of 2020), for serous adenocarcinoma of the ovary, grade 1-2-3 is not in use any more. It is either low grade or high grade serous ovarian carcinoma (LGSOC or HGSOC). LGSOC and HGSOC do not represent 2 differentiations of 1 cancer but rather 2 completely different cancer types, comparable to clear cell, mucinous, endometrioid,… This is also important as in several of your serous carcinomas, you do not find a TP53 mutation but only KRAS or PIK3CA: are these in fact LGSOC according to the current WHO guidelines?

Another strange fact is the serous borderline tumor that has received 6 lines of chemotherapy, and leading to the death of the patient. So this is not a borderline tumor! It might started as a borderline tumor, but it must have evolved into a LGSOC – in the WHO guidelines, invasive peritoneal implants are now regarded as LGSOC.

Please correct your numbers and use more logical denominators, or be more consequent.

- Lines 89-90: in seven samples no mutation was found à the abstract mentions 8 (line 24: in 36/44 at least one mutation… was detected) and in line 77 it also says 36 patients.

- Lines 27-28 Median time from PTB to TT start was 14 days. First, on line 133 it says 18 days. Second, these 14 days (or 18, whatever is correct) are very misleading. Both in the results (line 133-134) and discussion (207-208), you do mention the time period that really matters to the clinicians: median time between indication for PTB and TT was 65 days: when a patient presents with recurrence and an alternative treatment should be proposed, genomic profiling with PTB still needs to be performed.  This delay of 65 days is more than 2 months, which is reality, but also an important explanation for the poor performance status of ovarian cancer patients. And maybe that conclusion on lines 210-211 should be more emphasized!

- Lines 26 and 104: in 31 patients (70%) a TT was proposed. The 70% is based on the total amount of 44 patients, but it would be more logic to use only the 36 patients with a possible molecular or immunohistochemical target (moreover because in 2 patients no IHC was performed, so that would have been 42 patients). Also line 27: instead of 12/44 us the 31 patients (for whom TT was recommended) as denominator because that is more relevant: 12/31 (see also line 109, where you do explain that 12 patients is only 27% of the total study population). In line 30 you do use the denominator of 31 patients: in 61% TT was not administered = 19/31 (so you’re not very consequent in choosing your relevant denominators).

- Lines 104-107: In 31 patients a TT was proposed based on mutations or IHC: what is the proportion: how many are based on IHC? And how many of the final 12 were based on IHC?

Minor remarks on PD-L1. First E1L3N is a poor antibody as it gives too many false or high positivity. Also trials with pembro, atezo, nivolumab are (sort of) validated with other antibodies – of note: I have no commercial interests. Line 101-102: it is combined positive score and not combined score. Minor remark on abbreviations: please mention the full name the first time of use. Part of the confusion arises because the M&M section has been placed at the end. (Line 28 and 135: CBR, line 111: ICI; line 113, TTF; line 115: CR, PR, SD). Consider the use of a graphic to represent TTF, with a horizontal axis showing the duration since start of treatment and the vertical axis the individual 12 patients.

Reviewer 2 Report

There are strong evidences that the gene sequencing techniques support the comprehensive new cancer treatment. It would be chance for prolonged life of ovarian cancer patients with inoperable or metastatic cancers, and those for whom the standard treatment has failed. The Authors took on the challenge of the clinical applicability of PTB for ovarian cancer patients. It is a valuable work, especially, that there is little data in this field, and further extensive studies are required due to high mortality among OC patients.

However, this work requires revision and some corrections.

 Introduction: 

The introduction could be described in more detail, with some examples related to molecular aberrations relevant for the treatment of cancer.

 The last sentence starting with the phrase "To achieve this purpose, we analyzed the clinical data of all ovarian cancer patients ..." should be clarified: what kind of analysis the Authors meant in their research, what were their assumptions. The goal of the study is poorly explained. In a well-written discussion there are a few sentences justifying the study. I suggest moving some of them to the Introduction.

 Results:

 A relatively small number of ovarian cancer patients have been included into the study. However, this is justified for short-term (5 years) and single-centre studies.

Selection criteria of histopathological markers and genes should be justified and described widely.

What were the criteria for matching OC patients with a given gene mutation for the treatment group?

Line 64: From the statistical point of view, it is hard to call a "population" a group of 44 patients. It should be stated "group" instead of "population".

Language:

English should be corrected due to some typing errors, grammar and stylistic awkwardness.